# Thawing of Frozen Hairtail (*Trichiurus lepturus*) with Graphene Nanoparticles Combined with Radio Frequency: Variations in Protein Aggregation, Structural Characteristics, and Stability

**DOI:** 10.3390/foods13111632

**Published:** 2024-05-24

**Authors:** Fang Tian, Wenyuchu Chen, Xiaohan Gu, Weiliang Guan, Luyun Cai

**Affiliations:** 1Key Laboratory of Health Risk Factors for Seafood of Zhejiang Province, School of Food and Pharmacy, Zhejiang Ocean University, Zhoushan 316022, China; tianfang@zjou.edu.cn (F.T.); chenwenyuchu@zjou.edu.cn (W.C.); jy6214gxh@163.com (X.G.); 2Ningbo Innovation Center, College of Biosystems Engineering and Food Science, Zhejiang University, Ningbo 315100, China; 3School of Light Industry and Food Engineering, Guangxi University, Nanning 530004, China; 4School of Chemical and Biological Engineering, NingboTech University, Ningbo 315100, China

**Keywords:** graphene nanoparticles, radio-frequency thawing, hairtail, protein structure, texture

## Abstract

Efficient thawing can preserve the quality of frozen hairtail (*Trichiurus lepturus*) close to that of fresh hairtail. In contrast to air thawing (AT) and radio-frequency thawing (RT), this study looked at how graphene oxide (GO) and graphene magnetic (GM) nanoparticles paired with RT affect the microstructure and protein conformation of hairtails after thawing. The results suggested that GM-RT can reduce the myofibrillar protein (MP) damage and be more effective than other thawing treatments, like AT, RT, and GO-RT, in maintaining the microstructure of hairtail. The particle size and zeta potential showed that GM-RT could reduce the aggregation of MP during the thawing process compared to other thawing methods. Moreover, the texture of the hairtail after GM-RT exhibited higher hardness (1185.25 g), elasticity (2.25 mm), and chewiness (5.75 mJ) values compared to other thawing treatments. Especially compared with RT, the GM-RT treatment displayed significant improvements in hardness (27.24%), a considerable increase in springiness (92.23%), and an increase in chewiness (57.96%). GO-RT and GM-RT significantly reduced the centrifugal loss. The scanning electron microscopy results demonstrated that the effect of GM-RT was more akin to that of a fresh sample (FS) and characterized by a well-organized microstructure. In conclusion, GM-RT effectively diminished the MP aggregation and improved the texture of thawed fish. It can be regarded as a viable alternative thawing technique to enhance MP stability, which is vital for preserving meat quality.

## 1. Introduction

The hairtail (*Trichiurus lepturus*) found in the East China Sea is a popular marine fish with significant production levels in China. It is widely consumed for its distinctive flavor and nutritional value [1]. However, hairtail is rich in protein and unsaturated fatty acids, which are degraded by microbial activity, leading to the short shelf life of hairtail [2]. Freezing storage is one of the critical methods for preserving and long-term storing aquatic products, which involves lowering the temperature below freezing point to alleviate microbial activity. Thawing, an essential process before secondary processing or consumption of frozen products, is critical for maintaining the flesh quality. The traditional thawing methods, such as air thawing and water-immersion thawing, often need a longer time, measured in hours, and are prone to resulting TBARs increasing, which means the lipid oxidation in the fish becomes severe and the protein gradually denatures in frozen fish [3]. In addition, the degree of protein degeneration is the critical factor affecting the texture of fish [4]. Moreover, hairtail varies in thickness, so traditional thawing techniques may cause frozen products to be overcooked. Rapid surface thawing causes proteins to coagulate and form a hard shell, hindering the internal temperature from rising simultaneously [1]. Therefore, an effective and edible thawing method should be considered to improve the frozen fish’s texture, water-holding capacity, and protein physicochemical properties while preventing structural damage to hairtail fish caused by uneven heating.

Several new thawing technologies and their combinations have emerged in recent years. They have been developed and are widely used in industrial processes to maintain the quality of fish and shorten the thawing time. For example, microwave thawing can provide frozen mackerel with better thermal stability, more stable protein conformation, and lower protein oxidation [5]. Cai et al. reported that far-infrared thawing could improve the texture properties and freshness more than air thawing [6]. Ultrasonic is commonly used for thawing to enhance the quality of meat, characterized by a high thawing speed and uniform thawing [7]. However, these techniques exhibit disadvantages, including time-intensive processes, inconsistent thawing, and the potential to cause irreversible physical or chemical harm to fish products [8]. Therefore, some studies have reported that combined thawing methods could address the above problems. Different combinations of far-infrared thawing were used to make the MP of the fish’s secondary and tertiary structures more stable, including visible/near-infrared plus far-infrared thawing and magnetic nanoparticles plus far-infrared thawing [9,10]. Ultrasonic thawing combined with far-infrared thawing, dual-frequency ultrasonic thawing, and ultrasonic-assisted low-temperature (U-LT) thawing can decelerate the oxidation of lipids and proteins in frozen products [8,11,12]. In this context, applying integrated thawing strategies is essential to retain the fish’s nutritional attributes and texture after storage in frozen conditions.

Radio-frequency (RF) heating, as one of the efficient thawing and tempering methods, has attracted increasing attention [13]. This thawing technology mainly relies on the heat generated by the RF electromagnetic field to heat frozen objects [14]. Compared with traditional heating methods such as hot-air or water-bath heating, the advantage of RF thawing is fast and large-volume heating because RF waves can penetrate inside and outside the item. However, uneven thawing and surface overheating usually appear during the RF thawing process, causing irreversible physical or chemical damage to fish products [15]. GO and GM represent two unique derivatives of graphene. GO, an oxidized form of graphene, gains hydrophilicity and reactivity due to functional groups from oxidation disrupting its conjugated system [16]. Conversely, graphene magnetic, as a nanocomposite material, merges graphene with magnetic substances (like magnetic metal particles), combining graphene’s high surface area for uniform heating with the ability of magnetic material to generate heat when it is subjected to an alternating electromagnetic field [17]. In addition, low doses of magnetic nanoparticles do not harm humans and have also been approved by the United States Food and Drug Administration (FDA) [18]. Using a water-based dispersion of graphene nanoparticles as a thawing medium minimizes the need for specific frozen product shapes and effectively reduces uneven heating or overheating in different meat parts. Nevertheless, the effect of combining GM and GO with RF to thaw out situations of MP aggregation and the texture changes in hairtail has not been studied.

Our previous study showed that GO-RT and GM-RT can successfully reduce the thawing time and attenuate lipid and protein oxidation. We also need to reveal the aggregation of MP and the texture change in the hairtail, which can validate the changes in hairtail quality due to different thawing techniques. In this study, we introduced the combination of GM and GO nanoparticles with RF thawing to achieve fast and uniform defrosting of frozen hairtail. Specifically, particle size, microstructure, and texture changes were carried out to evaluate the effects of different thawing methods (AT, RT, GO-RT, and GM-RT). Meanwhile, AT and RT methods were employed for reference. This experiment investigated the effects of simultaneous radio-frequency thawing and graphene nanoparticles on MP aggregation and hairtail texture.

## 2. Materials and Methods

### 2.1. Sample Preparation

Each frozen hairtail (*Trichiurus lepturus*) was approximately 60 cm long and weighed 0.5 kg. The fish were purchased from the local market in Zhoushan, Zhejiang Province, China. The hairtails were packed in an icebox and transferred to the laboratory directly. Upon arrival, the samples were packed in 2 × 2 × 0.5 cm^3^ in a refrigerator at −80 °C for 3 days and ensured to be completely frozen for further experiments [19]. The aqueous dispersion of graphene oxide nanoparticles and the aqueous dispersion of magnetic graphene oxide nanoparticles were purchased from Jiangsu Xianfeng Nanomaterial Technology in Nanjing (China) Co., Ltd. The radio-frequency instrument (Dotwil Intelligent Technology in Shanghai (China) Co., Ltd., Shanghai, China) was used to thaw the frozen samples at a 40.68 MHz frequency.

### 2.2. Experimental Setup

Frozen samples measuring 2 × 2 × 0.5 cm^3^ were thawed using air thawing (AT), radio-frequency thawing (RT), oxidized graphene nanoparticles combined with radio-frequency thawing (GO-RT), and magnetic graphene nanoparticles combined with radio-frequency thawing (GM-RT). GO and GM nanoparticles are used as aqueous dispersions, and their concentrations were 0.1 mg mL^−1^. One piece of frozen hairtail was placed in a polyethylene bag (Tupu Daily Chemicals (China), Co., Ltd., Wuxi, China). The gas in the bag was removed manually to ensure the surface contact between sample and medium before heat sealing (Tupu Daily Chemicals in Shanghai (China) Co., Ltd.). The frozen samples were thawed using a radio-frequency thawing device (provided by Shanghai Dian Wei Intelligent Technology in Shanghai Co., Ltd., China) at a frequency of 40.68 MHz [20]. When the center temperature of a sample reached 4 ± 1 °C, it was considered completely thawed. The probe thermometer was purchased from Deli Group in Ningbo (China) Co., Ltd. The times of the different thawing technologies (AT:150 min, RT: 3 min, GO-RT: 3 min, and GM-RT: 3 min) were confirmed in pilot tests.

#### 2.2.1. AT

The frozen hairtails were packaged in polyethylene bags and thawed by air at 15 °C for 150 min.

#### 2.2.2. RT

The frozen hairtails were packaged in polyethylene bags and thawed by a radio-frequency thawing device at 15 °C for 3 min.

#### 2.2.3. GO-RT

The frozen hairtails were packaged in polyethylene bags and thoroughly soaked in the GO nanoparticles solution (0.1 mg/mL) at a mass ratio of 1:3 for 1 h at 4 °C and thawed by a radio-frequency thawing device for 3 min.

#### 2.2.4. GM-RT

The frozen hairtails were packaged in polyethylene bags and thoroughly soaked in the GM nanoparticles solution (0.1 mg/mL) at a mass ratio of 1:3 and thawed by a radio-frequency thawing device for 3 min.

### 2.3. Texture Analysis

Appropriate modifications were made according to the method of Zhu et al. [20]. The hairtail meat samples measuring 2 × 2 × 0.5 cm^3^ were measured at 4 °C. The measurement used a Brookfield CT3 (BROOKFIELD in Middleboro, Massachusetts (USA) Co., Inc.) with a TA 25/1000 cylindrical probe. The single compression cycle test mode was employed, with the probe compression direction perpendicular to the direction of the fish’s muscle fiber. The measurement parameters were: test speed of 30 mm/min, compression distance of 4 mm, and trigger force of 4.5 g.

### 2.4. Centrifugal Loss

A sample of minced, thawed hairtail dorsal muscle packed into two sheets of filter paper was centrifuged at 6000× *g* for 10 min at 4 °C. The sample was taken out and weighed. The centrifugal loss was calculated using the following equation:$$Centrifugal\;loss=\frac{\left(M2-M3\right)}{M2}\times 100\%$$

M2: the weight of the thawed sample, M3: the weight of the sample after centrifugation.

### 2.5. Scanning Electron Microscope

The specimens were sectioned into cubes measuring 3 × 3 × 3 mm^3^ and immersed in a 2.5% glutaraldehyde solution inside a 4 °C refrigerator to stabilize the tissue for 24 h. They were subsequently rinsed with phosphate buffer (0.2 M, pH 7.2) for 20 min. They were rinsed with deionized water to remove any fixative residues. The specimens underwent a dehydration process using alcohol concentrations of 50%, 70%, 80%, 90%, and 95%, with each stage lasting 15 min. Following these steps, the sections were lyophilized in a freeze-dryer for 48 h. The microstructural details of the samples were analyzed with a scanning electron microscope operating at 15.0 kV [21].

### 2.6. Extraction of MP

Appropriate modifications were made according to the method of Lefevre et al. [22]. A total of 3 g of thawed fish meat were homogenized with 30 mL of buffer A at 13,000 r/min for 60 s. The mixture was centrifuged at 8000 r/min for 10 min at 4 °C, and the supernatant was discarded. The precipitated fraction was collected and added to 15 mL of buffer A, followed by thorough mixing. After washing it twice, the precipitated fraction was collected again. The residue was then added to 15 mL of buffer B, homogenized at 13,000 r/min for 60 s, fully dissolved, and left on ice for 2 h. The solution was filtered through double-layer gauze, resulting in the obtained myofibril solution. (Buffer A: 20 mmol/L phosphate buffer containing 100 mmol/L NaCl, 1 mmol/L EDTA, pH 7.0, Buffer B: 25 mmol/L phosphate buffer containing 0.6 mmol/L NaCl, pH 7.0.)

### 2.7. Measurement of Particle Size

Appropriate modifications were made according to the method of Zhang et al. [8]. Protein aggregation was measured with a laser particle size analyzer. Absorption was set to 0.001°. Myogenic fibronectin solution was diluted to 4 mg/mL with phosphate buffer (20 mM, 100 mM NaCl, 1 mM EDTA, pH 7.0) and placed in a specific cuvette. The samples were recorded using Zeta sizer software (2.3.0.62) to record the intensity profile and the average particle size. The temperature was set at 25 °C, the equilibration time and scanning time were 10 s, and the average number of readings per sample was 5. The average measurement time was approximately 120 s. All the samples were analyzed in triplicate, and the readings were expressed as mV.

### 2.8. Measurement of Zeta Potential

The zeta potential of myogenic fibronectin was measured using a nanoparticle size analyzer [23]. The concentration of the myogenic fibronectin solution was adjusted to 0.5 mmol/mL with buffer B to avoid multiple scattering effects and operated at 25 °C and 23 V/cm electric field strength.

### 2.9. X-ray Powder Diffraction

The samples were analyzed in triplicate using an X-ray diffractometer under the following conditions: X-ray tube, Cu–Ka (nickel filter); 40 kV; voltage, 40 mA; scanning form, 2θ of 5–40°; scanning speed, 8°/min; and scanning wavelength, 0.154 nm [24].

### 2.10. Fluorescence Spectroscopy

The intrinsic fluorescence spectrum of the untreated and treated samples was measured using a fluorescence spectrophotometer, according to a previous study [10]. A blank control group without MP was prepared using 0.1 mol/L PBS-C buffer, and the detector sensitivity was set to automatic, with a bandwidth of 2 nm and an excitation wavelength fixed at 292 nm. Scanning was conducted in the range of 300–500 nm.

### 2.11. Differential Scanning Calorimetry

The thermal transitions in the whole-muscle specimens were analyzed with a differential scanning calorimeter. The device (Erich NETZSCH B.V. & Co. Holding KG in Selb, Germany) was temperature-calibrated using an Indium reference thermogram. A 0.2 g meat sample was enclosed in a stainless-steel pan, sealed, and heated from 20 to 100 °C at a rate of 0.5 °C per minute. A reference scan was performed using an empty pan [25]. The software Universal Analysis (2000, Version 4.5A, TA Instruments, Stuttgart, Germany) was utilized to determine the peak transition temperature (T_max_) based on the thermogram.

### 2.12. Statistical Analysis

All the data for each treatment are represented as the mean of three replicates. A one-way analysis of variance (ANOVA) and Duncan’s multiple range were carried out using SPSS 22.0 statistical software (SPSS Inc., Chicago, IL, USA), and the difference was considered significant at 5% (*p* < 0.05). All the data are represented as the mean ± standard deviation (SD). All the figures were plotted using Origin Pro 14.0 (Origin Lab Co., Northampton, MA, USA).

## 3. Results and Discussion

### 3.1. Effects of Different Thawing Methods on the Particle Size and Zeta Potential of Hairtail MP 

The particle size is a reliable approach during the thawing process, and it can evaluate the degree of protein aggregation under oxidative attack [26]. The peak intensities of the samples and the particle sizes under different thawing methods are shown in Figure 1A,B. The particle sizes of all the samples ranged from 100 to 10,000 nm. Compared with FS, the different thawing treatments resulted in a significant increase in the particle size of the MP samples, indicating varying degrees of aggregation during the thawing process (*p* < 0.05). In this part, the MP observed in GO-RT and GM-RT had a smaller particle size range than in AT and RT. As the peak size increased, the distribution narrowed, and the particle size became more uniform. The results showed some degree of aggregation in the AT and RT samples and slighter aggregation in the GM-RT and GO-RT samples. The increase in particle size might be due to the formation of aggregates and the oxidative denaturation of the MP, especially thermal damage to hydrogen or electrostatic bonds in the protein structure. Due to this thermal denaturation, the hydrophobic residues on the protein surface were exposed [27]. The reaction of carbonyl groups with free amino groups formed disulfide-bonded di-tyrosine cross-links between the protein molecules, which tended to form protein aggregates [28]. This was consistent with the experimental results and similar to our team’s previous research results [8]. In addition, external factors such as shock, pH, dehydration, and oxidation reactions also played an important role [29]. These results indicated that GM-RT and GO-RT might result in a better improvement in these unfavorable factors and reduce the MP aggregation.

The stability of the MP was directly correlated with the absolute ζ value, and the zeta potential was a significant predictor of the distributed system stability [30]. When the charges of the protein surface were decreased, the absolute value of ζ (|ζ|) would become smaller, which indicated that the protein molecules were unfolded, leading to a decrease in the electrostatic repulsion [29]. As shown in Figure 1C, all the samples with different thawing treatments showed a negative ζ value. There was a decreased phenomenon of the |ζ| values in thawed samples compared to FS, showing that the MP was damaged during thawing. The ζ value for RT was 3.01 ± 0.72 and indicated the low stability of the MP. The ζ values for GO-RT and GM-RT were 3.57 ± 0.21 and 3.59 ± 0.2, respectively, compared with RT, indicating the good stability of the MP. The improved ζ value may be because the isotropic charge on the protein surface increased, and the mutual repulsion of more isotropic charges makes the protein solution more stable [6]. Consequently, the results concerning the particle size and zeta potential suggested that GM-RT could reduce the aggregation and maintain the stability of the protein during the thawing process.

### 3.2. Effects of Different Thawing on MP Secondary and Tertiary Structures in Hairtail

X-ray powder diffraction (XRD) is an analytical technique used to identify crystal structures, detecting intra- and inter-molecular interactions within polymer networks [31]. The peaks on the spectrum indicate the diffraction signals from different crystal planes in the sample. The position of these peaks (2-Theta angle) and their intensity (intensity counts) can be used to identify and compare the crystalline structures of samples. As shown in Figure 2, for all the groups, the three prominent crystalline peaks were primarily located around 32.1°, 46.2°, and 56.8°. All the groups exhibited a similar broad peak corresponding to the amorphous state, confirming the amorphous structure of myogenic fibrin. Compared to the FS group, the samples treated with AT, RT, GO-RT, and GM-RT showed increased peak intensities at 32.1°, 46.2°, and 56.8°, indicating that the thawing treatments caused myosin aggregation. Moreover, the peak intensities of the RT and AT groups rose to higher values, suggesting that these treatments disrupt the integrity of the MP. In contrast, the peak of the GM-RT was closer to the FS group. These results indicated that GM-RT caused less damage to the MP and resulted in more minor structural changes. These findings were also consistent with previous experiments [32].

The intrinsic fluorescence spectrum, by reflecting changes in the tryptophan (Trp) residues, could serve as an evaluation indicator of the tertiary structure of MPs [33]. The intrinsic fluorescence of the Trp residues was particularly sensitive to the microenvironment’s polarity and was an essential technical tool for detecting the tertiary conformation of the proteins. As shown in Figure 2B, for different thawing at the fluorescence intensity (FI) of RT, AT, FS, GO-RT, and GM-RT were the highest to the lowest. RT and AT were higher in fluorescence intensity than FS, indicating that most of the Trp residues were buried in the core of the protein [18]. On the contrary, the fluorescence intensity decreased in GM-RT and GO-RT, possibly due to the effect of the Trp in the solvent or oxidation in the partially or fully swollen state, where they were exposed to the solvent in the solvent or oxidation [21]. The magnitude of the maximum fluorescence emission peak (λ_max_) reflected the extent of the conformational change in the protein. It has been well established that the larger the redshift, the greater the conformational change in the protein during denaturation [34]. As shown in Table 1 and Figure 2B, compared to the FS samples, the shift in the λ_max_ was slightly longer (red-shifted) for all the other samples, which indicated that the microenvironment of the Trp residues shifted to a polar environment due to unfolding. However, the difference was insignificant regarding the maximum fluorescence emission wavelength (*p* > 0.05). The results indicated that the four thawing methods had no significant effect on the tertiary structure of the proteins compared to that of the fresh samples.

### 3.3. Effects of Different Thawing on Microstructure in Hairtail

Different thawing treatments are critical for MP aggregation and hairtails with a stable and complete structure. As shown in Figure 3, the muscle fibers were apparent and neatly arranged in an orderly manner. The myofibrillar units and myogenic fibers of the reticular structure could be observed in the FS. When using different thawing treatments, the microstructure was destroyed. In contrast, compared with the other samples, the microstructure of the AT sample was very blurred, and it was challenging to identify regular myogenic fibers. During the thawing, the MP denaturation and endocardial rupture may destroy the muscle fiber structure [17]. On the contrary, GO-RT and GM-RT showed much more compact and orderly microstructures, with more stable protein conformation, and the hardness of GO-RT and GM-RT improved by 12.88% and 27.24% compared with AT. These results were likely due to the magnetic graphene nanoparticles avoiding an excessive heating temperature, successfully causing structural damage. In particular, the GM-RT samples had the straightest MP observed in the SEM images, which maintained the texture close to FS, as shown in Table 2.

### 3.4. Effects of Different Thawing Methods on Texture and Centrifugal Loss of Hairtail Muscle

Texture represents the tissue structure and sensory characteristics of aquatic products after undergoing the thawing process. As shown in Table 2, all the hardness, springiness, and chewiness showed a similar trend, reducing regardless of the thawing treatment. The results demonstrated that the highest hardness (1475.5 g), springiness (2.37 mm), and chewiness (6.22 mJ) were found in fresh hairtail. In comparison with FS, the hardness (g), springiness (mm), and chewiness (mJ) of both the GM-RT and GO-RT groups showed relatively minor variations, which agreed with the changes in the hairtail microscope. AT and RT performed worse in terms of hardness, springiness, and chewiness, possibly due to the MP reaching the denaturation point of protein or the violent oscillation of hydrogen bond loosening [34]. When using GM-RT for thawing, these parameters were significantly increased by 27.2%, 92.3%, and 57.9%, respectively, making it the closest thawing method to the FS group. Gan found that the decrease in hardness was associated with the cross-linking, denaturation, and degradation of proteins during the thawing process [35]. The springiness was decreased in the AT and RT groups (*p* < 0.05). The reduction in hardness and springiness in both groups could be attributed to the uneven heating of the ice crystals formed after freezing, which damaged the tissue structure during thawing [36]. The scanning electron microscope image also confirmed this speculation. In conclusion, after thawing, the textural properties of all the samples decreased, but the GM-RT group had the best properties among all the thawing methods.

The centrifuge loss can indicate thawed samples’ water-holding capacity (WHC) [37]. As shown in Figure 4, the centrifugal losses for the AT (0.19 ± 0.03%) and RF (0.27 ± 0.02%) samples were significantly higher compared to the other samples (*p* < 0.05). This may be due to damage to the muscle protein (MP) structure during the thawing process, resulting in the exposure of hydrophobic groups [38]. On the other hand, the WHC was greatly enhanced by the GO-RT (0.09 ± 0.01%) and GM-RT (0.1 ± 0.01%) approaches, which could be attributable to their ability to evenly heat frozen hairtail. As a result, the centrifugal loss was decreased, and the WHC was improved due to the preservation of MP integrity [39]. These results agreed with the previously described texture analysis and scanning electron microscopy results.

### 3.5. Differential Scanning Calorimetry

DSC is utilized to assess the alterations in protein denaturation and aggregation, with the primary indicator of protein structure instability being the reduction in the transition temperature [40]. Meanwhile, enthalpy changes (ΔH) were used to indicate the net content of the ordered secondary structure [41]. Both of them were negatively correlated with the degree of protein denaturation. The effect of the different thawing methods on the denaturation temperature and enthalpy values of fish samples is shown in Table 3, and the DSC thermograms of MP are shown in Figure 5. There are three major endothermic peaks in Figure 5; according to previous research studies, peak 1 represents the thermal change caused by the denaturation of the myosin head, peak 2 represents the thermal change of the myosin tail and sarcoplasmic proteins, and peak 3 represents the thermal change of actin [42]. The results showed that the T_max1_ in the GM-RT group was higher than in the other samples, indicating that the myosin head was more stabilized in the RT-Mag group. Although the T_max 1_ of FS was lower than that of GO-RT and GM-RT, and no significant difference was detected (*p* > 0.05). The T_max1_ of AT and RT was reduced during the thawing process, indicating both thawing treatments adversely impacted the partial denaturation of the myosin head in hairtail. As for AT and RT, the protein denaturation might be related to overheating, which led to the loosening of the hydrogen bonds and reduced the stability of the myosin head [6]. In peak 2 and peak 3, the difference in the T_max2_ and T_max3_ was not significant (*p* > 0.05) for the five groups, indicating that myosin, sarcoplasmic proteins, and actin exhibit strong stability under the thawing treatments [10]. Based on the above results, we found that myosin, sarcoplasmic proteins, and actin had good thermal stability during thawing. In contrast, the myosin head’s stability depended on the different thawing treatments. Among the four thawing techniques, GM-RT showed better thermal stability.

## 4. Conclusions

This study demonstrated that four thawing methods (AT, RT, GO-RT, GM-RT) affected frozen hairtail regarding the quality characteristics and MP aggregation. Based on assessments of the data mentioned above, the particle size, zeta potential, and secondary and tertiary structures of MP thawed by GM-RT and GO-RT were better than those of the other treatments, and both of them showed a more stable state. At the same time, compared with the other thawing treatments, the GM-RT texture and microstructure were kept closest to the FS. Regarding the water-holding capacity, the GO-RT and GM-RT samples had lower centrifugal loss; DSC analysis also showed that GM-RT could keep the MP head stable to reduce the oxidation and damage caused by thawing. We found that during the thawing process, graphene oxide (GO) and graphene magnetic (GM) nanoparticles were uniformly dispersed on the hairtail surface, preventing sample overheating and ensuring uniform heating. This minimized the MP structural damage and self-aggregation. In summary, GM-RT, as a new thawing treatment, could effectively maintain the quality of frozen hairtail and improve its MP aggregation situation. However, there are still some limitations, including the use of magnetic graphene nanoparticles, which may increase the cost of the thawing process. We will conduct further studies to determine the optimal processing settings for magnetic graphene (MG) nanoparticles during thawing. This research introduced some novel thawing technologies for application in the food industry.

## Figures and Tables

**Figure 1 foods-13-01632-f001:**
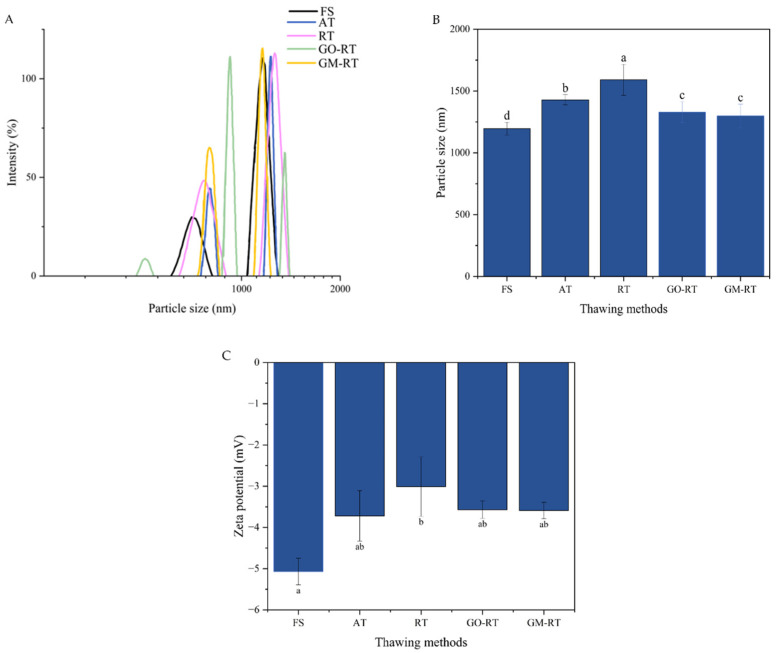
Particle diameter distribution (**A**), particle size (**B**), and zeta potential (**C**) of MP in hairtail thawed under different methods. The “a–d” letters indicate significant differences (*p* < 0.05). The error bar was the standard deviation. FS: Fresh sample; AT: Air thawing; RT: Radio-frequency thawing; GO-RT: Oxidized graphene nanoparticles combined with radio-frequency thawing; GM-RT: Magnetic graphene nanoparticles combined with radio-frequency thawing.

**Figure 2 foods-13-01632-f002:**
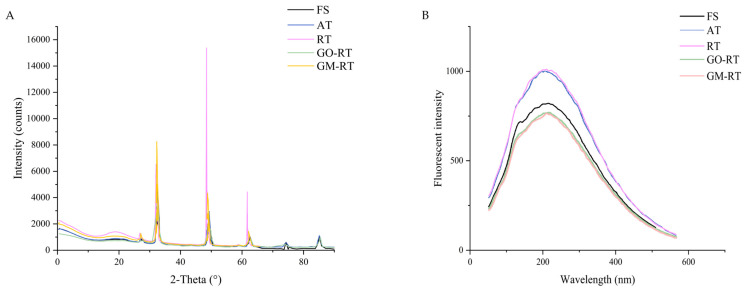
X-ray powder diffraction (**A**) and fluorescence spectroscopy (**B**) in hairtail after different thawing methods. FS: Fresh sample; AT: Air thawing; RT: Radio-frequency thawing; GO-RT: Oxidized graphene nanoparticles combined with radio-frequency thawing; GM-RT: Magnetic graphene nanoparticles combined with radio-frequency thawing.

**Figure 3 foods-13-01632-f003:**
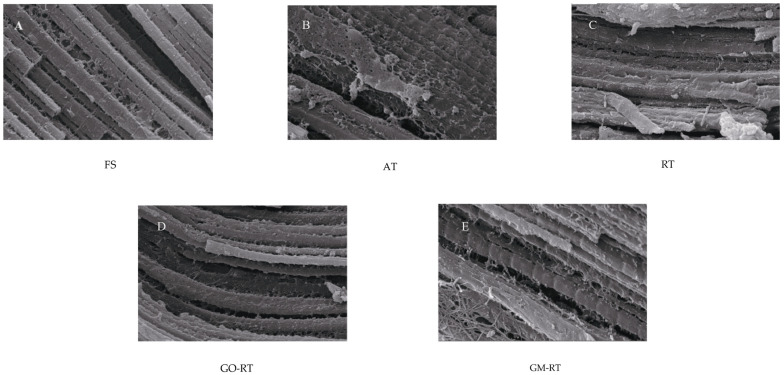
Scanning electron microscope image of hairtail under different thawing methods (×1000, FS: Fresh sample; AT: Air thawing; RT: Radio-frequency thawing; GO-RT: Oxidized graphene nanoparticles combined with radio-frequency thawing; GM-RT: Magnetic graphene nanoparticles combined with radio-frequency thawing.

**Figure 4 foods-13-01632-f004:**
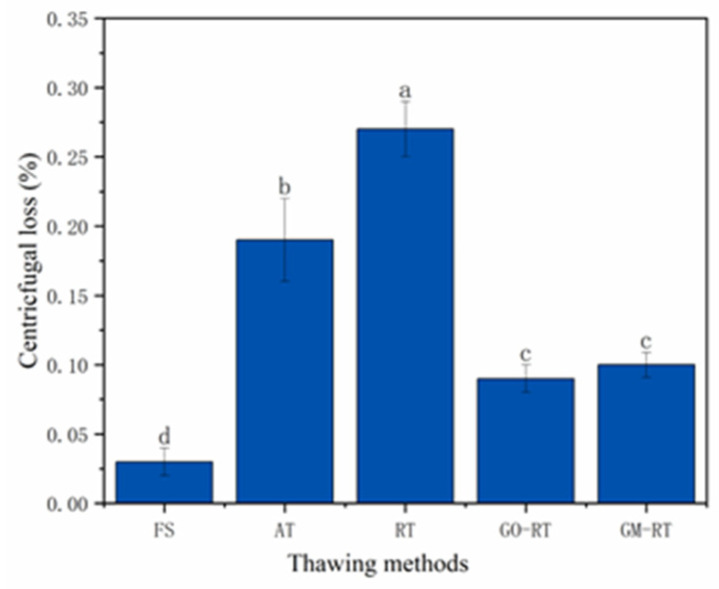
Centrifugal loss of samples as affected by different thawing methods. FS: Fresh sample; AT: Air thawing; RT: Radio-frequency thawing; GO-RT: Oxidized graphene nanoparticles combined with radio-frequency thawing; GM-RT: Magnetic graphene nanoparticles combined with radio-frequency thawing. The “a–d” letters in the same column indicate significant differences (*p* < 0.05).

**Figure 5 foods-13-01632-f005:**
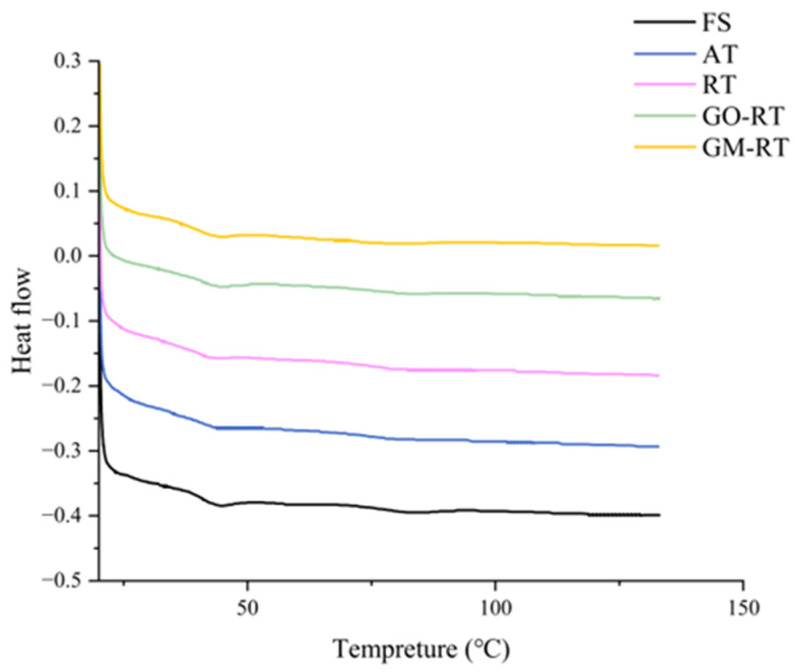
DSC scanning curve of hairtail under different thawing methods. FS: Fresh sample; AT: Air thawing; RT: Radio-frequency thawing; GO-RT: Oxidized graphene nanoparticles combined with radio-frequency thawing; GM-RT: Magnetic graphene nanoparticles combined with radio-frequency thawing.

**Table 1 foods-13-01632-t001:** Changes in the maximum fluorescence emission peak and fluorescence intensity of myofibrillary protein in hairtail under different thawing treatments.

	λ_max_	FI
FS	297.67 ± 4.02 ^a^	789.23 ± 44.30 ^ab^
AT	301.01 ± 6.61 ^a^	933.12 ± 74.83 ^a^
RT	302.14 ± 6.23 ^a^	943.11 ± 98.33 ^a^
GO-RT	300.04 ± 3.12 ^a^	779.15 ± 53.63 ^ab^
GM-RT	299.87 ± 2.11 ^a^	780.09 ± 43.18 ^ab^

Different letters in the same column indicate significant differences (*p* < 0.05).

**Table 2 foods-13-01632-t002:** The changes in the texture of hairtail meat under different thawing methods.

	Hardness (g)	Springiness (mm)	Chewiness (mJ)
FS	1475.5 ± 175.1 ^a^	2.37 ± 0.26 ^a^	6.22 ± 1.51 ^ab^
AT	1065.2 ± 83.5 ^ab^	1.53 ± 0.43 ^b^	4.59 ± 1.91 ^b^
RT	931.5 ± 76.5 ^b^	1.17 ± 0.55 ^b^	3.64 ± 2.38 ^b^
GO-RT	1179.9 ± 66.5 ^ab^	2.01 ± 0.35 ^a^	5.71 ± 2.63 ^a^
GM-RT	1185.25 ± 131.8 ^ab^	2.25 ± 1.33 ^a^	5.75 ± 2.26 ^ab^

Different letters in the same column indicate significant differences (*p* < 0.05). FS: Fresh sample; RT: Radio-frequency thawing; GO-RT: Oxidized graphene nanoparticles combined with radio-frequency thawing; GM-RT: Magnetic graphene nanoparticles combined with radio-frequency thawing.

**Table 3 foods-13-01632-t003:** T_max_ and ΔH values obtained under different thawing methods for hairtail muscle.

	Peak1		Peak2		Peak3	
	T_max1_	ΔH_1_	T_max2_	ΔH_2_	T_max3_	ΔH_3_
FS	44.23 ± 0.02 ^ab^	0.92 ± 0.09 ^a^	59.60 ± 0.04 ^a^	0.03 ± 0.01 ^a^	70.22 ± 0.61 ^a^	0.13 ± 0.04 ^ab^
AT	43.78 ± 0.66 ^b^	0.33 ± 0.87 ^a^	59.98 ± 0.13 ^a^	0.03 ± 0.19 ^a^	70.98 ± 0.98 ^a^	0.13 ± 0.97 ^ab^
RT	43.98 ± 0.23 ^b^	0.34 ± 0.09 ^a^	59.23 ± 0.10 ^a^	0.02 ± 0.01 ^a^	71.59 ± 0.78 ^a^	0.19 ± 0.02 ^a^
GO-RT	44.71 ± 0.22 ^a^	0.78 ± 0.53 ^a^	59.19 ± 0.09 ^a^	0.11 ± 0.21 ^a^	71.23 ± 0.43 ^a^	0.14 ± 0.02 ^ab^
GM-RT	44.62 ± 0.21 ^a^	0.83 ± 0.18 ^a^	59.21 ± 0.11 ^a^	0.11 ± 0.13 ^a^	71.65 ± 0.33 ^a^	0.13 ± 0.03 ^ab^

Different letters in the same column indicate significant differences (*p* < 0.05). FS: Fresh sample; RT: Radio-frequency thawing; GO-RT: Oxidized graphene nanoparticles combined with radio-frequency thawing; GM-RT: Magnetic graphene nanoparticles combined with radio-frequency thawing.

## Data Availability

The original contributions presented in the study are included in the article, further inquiries can be directed to the corresponding author.

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
