# Peer review of "Thawing of Frozen Hairtail (Trichiurus lepturus) with Graphene Nanoparticles Combined with Radio Frequency: Variations in Protein Aggregation, Structural Characteristics, and Stability"

_foods, 2024, doi:10.3390/foods13111632_

Round 1

Reviewer 1 Report

Comments and Suggestions for Authors

This study investigated the effects of using GO and GM combined with RT as novel thawing methods for Hairtail meat by measuring protein oxidation and aggregation. Overall, the quality of this manuscript is good. The experimental design is well-organized, and both the abstract and introduction are acceptable. The results are quite clear, and the discussion is well-developed, containing novelty as mentioned by the authors. There are only minor points I would like to address with the author. Here is my comment.

ABSTRACT

- Line27: “These results The Scanning” Please check this sentence.

- Line28: What’s “FS”, please provide the full name when mention as the first time.

- Line30: “which is important for realizing fast and uniform defrosting of frozen hairtail”. This conclusion is interesting. However, what’s parameter supported this hypothesis, since the experiment measured on the protein degradation/microstructure but without measure or monitor the ice crytal changes? Please clarify this point.

- Overall, the abstract is well-written. However, please make sure this is the first time for using GO/GM as thawing agent? This is because the author claimed as the new thawing method at the last sentence.

INTRODUCTION

- Line61-62: Author metioned to the change in secondary and tertiary structures of protein changes as various thawing processes. Why this study did not measure them, i.e. using FTIR, etc.?

- Line83: I suggested to provide the range of dossage of magnetic nanoparticles that can use in foods based on FDA standards.

MATERIAL AND METHODS

- Line100: Please provide the information how long to frozen sample before starting experiment.

- I suggest providing detailed information on the preparation of GM and GO, as they are new methods highlighted in this study. This will facilitate further research.

- Line111: For texture, why author used Bookfiled since this technique is suitable for semi-solid sample? Why author did not use TPA or shear force, which suitable for solid sample, particularly meat?

RESULTS AND DISCUSSION

- Why do GO and GM result in a more compact texture of meat after thawing? Additionally, why does GM, particularly as nanoparticles, appear to be more effective? Please include these points in the discussion.

- In the topic, author metioned to protein oxidation, what’s parameter directly referred to them?

- Furthermore, the alteration of the secondary and tertiary structure of proteins due to oxidation or aggregation appears to be a significant scientific aspect in describing the effectiveness of these thawing methods. If the author enhances the study by incorporating additional parameters or discussing them explicitly, it will enhance the scientific value for the readers.

Author Response

Dear editor,

Thank you very much for your letter and the comments on the manuscript (Manuscript ID foods-2964776 entitledEffects of different thawing methods based on graphene nanoparticles and radio frequency on reducing myofibrillar protein aggregation and improving the texture of hairtail. We have carefully considered the comments in preparing our revision. We submit the revised manuscript and our point-by-point responses here. All parts of the revised manuscript that were changed were marked in red. If you have any questions about this manuscript, please don’t hesitate to let me know.

Best regards,

Sincerely yours

Prof. Luyun Cai

Ningbo Research Institute, College of Biosystems Engineering and Food Science, Zhejiang University, Ningbo 315000, China

Reviewer#1

Comment 1: Line27: “These results The Scanning” Please check this sentence.

Response 1: We have revised the sentence in Line 27.

Comment 2: Line28: What’s “FS”, please provide the full name when mention as the first time.

Response 2: We have provided the full name of “FS” in Line 28.

Comment 3: Line30: “which is important for realizing fast and uniform defrosting of frozen hairtail”. This conclusion is interesting. However, what’s parameter supported this hypothesis, since the experiment measured on the protein degradation/microstructure but without measure or monitor the ice crytal changes? Please clarify this point.

Response 3: We have explored in our previous research to support this hypothesis, and we quoted the paper in Ref. [30] in Line 29.

Comment 4: Overall, the abstract is well-written. However, please make sure this is the first time for using GO/GM as thawing agent? This is because the author claimed as the new thawing method at the last sentence.

Response 4: We have modified this sentence in Line 30.

Comment 5: Line61-62: Author mentioned to the change in secondary and tertiary structures of protein changes as various thawing processes. Why this study did not measure them, i.e. using FTIR, etc.?

Response 5: We have used XRD and intrinsic fluorescence spectrum to express in secondary and tertiary structures of protein changes in Line 231.

Comment 6: I suggested to provide the range of dossage of magnetic nanoparticles that can use in foods based on FDA standards.

Response 6: We have agreed with the reviewer that more data could be useful to understand the dose of magnetic nanoparticles, but in our experiment, the nanoparticles didn’t enter the human body.

Comment 7: Please provide the information how long to frozen sample before starting experiment.

Response 7: We have added the relevant information in the text in Line 105.

Comment 8: I suggest providing detailed information on the preparation of GM and GO, as they are new methods highlighted in this study. This will facilitate further research.

Response 8: We have provided detailed information on the preparation of GM and GO in Line 109.

Comment 9: Line111: For texture, why author used Bookfiled since this technique is suitable for semi-solid sample? Why author did not use TPA or shear force, which suitable for solid sample, particularly meat?

Response 9: We have checked the instruction manual of Bookfiled CT3, which indicates that it can be used to detect TPA in solid samples in Line 131.

Comment 10: Why do GO and GM result in a more compact texture of meat after thawing? Additionally, why does GM, particularly as nanoparticles, appear to be more effective? Please include these points in the discussion.

Response 10: We have added  the reason in Line 360.

Comment 11: In the topic, author metioned to protein oxidation, what’s parameter directly referred to them?

Response 11: We have rewritten related information in Line 363.

Comment 12: Furthermore, the alteration of the secondary and tertiary structure of proteins due to oxidation or aggregation appears to be a significant scientific aspect in describing the effectiveness of these thawing methods. If the author enhances the study by incorporating additional parameters or discussing them explicitly, it will enhance the scientific value for the readers.

Response 12: We have used the position of peaks (2-Theta angle) and magnitude of the maximum fluorescence emission peak to reflect the changes of secondary and tertiary structure about protein.

Reviewer 2 Report

Comments and Suggestions for Authors

The manuscript, titled “Effects of Different Thawing Methods Based on Graphene Nanoparticles and Radio Frequency on Reducing Myofibrillar Protein Oxidation/Aggregation of Hairtail” describes the effects of thawing methods on the quality of hairtail fish. The authors implemented a novel thawing technique involving the incorporation of graphene oxide and graphene magnetic nanoparticles combined with radio frequency. They compared this innovative method with conventional air thawing and radio frequency techniques. The comparison was conducted by analyzing various aspects of the thawed products, including texture analysis, particle size, scanning electron microscope observation, evaluation of myofibrillar protein, measurement of zeta potential, X-ray diffraction, fluorescence spectroscopy, and differential scanning calorimetry (DSC). They discovered significant improvements with the use of graphene magnetic nanoparticles combined with radio frequency, suggesting it as a viable alternative for enhancing myofibrillar protein stability. This finding holds importance for achieving rapid and uniform defrosting of frozen hairtail.

This evaluation and innovation present promising advancements in the quality storage of frozen fish products. The experimental design appears sound, and the findings are interesting. However, the methods section requires further elaboration, and there are opportunities for improvement in the overall writing style. Certain sentences and statements lack adherence to scientific norms. Therefore, I recommend addressing the following issues to enhance the manuscript.

Some specific comments:

1.      Abstract: Line 16; The sentenceEfficient thawing can maintain the quality of frozen hairtail close to fresh hairtail (Trichiurus lepturus)” should be modified as “Efficient thawing can preserve the quality of frozen hairtail (Trichiurus lepturus) close to fresh one”.

2.   Abstract: Line 17-19; “This study aims to investigate graphene oxide (GO) and graphene magnetic (GM) nanoparticles combined with radio frequency (RT) in effecting protein conformation and microstructure of hairtail after thawing, compared with air thawing (AT) and RT”. The sentence needs modification.

3.      Abstract: Line 24; …… chewiness (5.75 mJ) should be ……… chewiness (5.75 mJ) values

4.      Abstract: Line 27; These results of…..

5.      Introduction: Lines 44; The authors did not perform any tests for lipid quality evaluation. So, I recommend to add any analysis for lipid oxidation tests.

6.      Introduction: Lines 58-59; how does traditional thawing cause overcooking? Please clarify…

7.      Introduction: Lines 46-47; “Nevertheless, these methods have shown drawbacks, such as being time-consuming, uneven thawing, and inducing irreversible physical or chemical damage to fish products”. Please add reference.

8.      Introduction: Line 90; the word “realize” is not suitable to express meaning….

9.      Introduction: Line 93-94; “The results obtained in this work will provide a theoretical basis for hairtail thawing treatment in the food industry” replace the sentence in conclusion.

10.  Sample Preparation: Line 97; Mention the length and weight of the fish.

11.  Sample Preparation: Line 100; Why did the authors freeze at –80 °C? Usual freezing is done at –40 °C.

12.  Experimental Setup: Line 103; The authors should state the source/ production device of oxidized graphene nanoparticles. They also require to add the instrumental model for a radio frequency thawing device.

13.  Name the device model and origin used to measure center temperature.

14.  Texture Analysis: Mention full specification of texture analyzer.

15.  Measurement of Particle Size: Add reference for the method used.

16.  Results: The Figure 2 & 3 need quality improvement for the readers.

17.  Conclusion: State the limitations (if any) and future research suggestions.

Comments on the Quality of English Language

The manuscript require overall language check by native/ professional person.

Author Response

Dear editor,

Thank you very much for your letter and the comments on the manuscript (Manuscript ID foods-2964776 entitledEffects of different thawing methods based on graphene nanoparticles and radio frequency on reducing myofibrillar protein aggregation and improving the texture of hairtail. We have carefully considered the comments in preparing our revision. We submit the revised manuscript and our point-by-point responses here. All parts of the revised manuscript that were changed were marked in red. If you have any questions about this manuscript, please don’t hesitate to let me know.

Best regards,

Sincerely yours

Prof. Luyun Cai

Ningbo Research Institute, College of Biosystems Engineering and Food Science, Zhejiang University, Ningbo 315000, China

Reviewer#1

Comment 1: Abstract: Line 16; The sentence “Efficient thawing can maintain the quality of frozen hairtail close to fresh hairtail (Trichiurus lepturus)” should be modified as “Efficient thawing can preserve the quality of frozen hairtail (Trichiurus lepturus) close to fresh one”.

Response 1: We have revised the manuscript to the language and grammar in Line 17.

Comment 2: Abstract: Line 17-19; “This study aims to investigate graphene oxide (GO) and graphene magnetic (GM) nanoparticles combined with radio frequency (RT) in effecting protein conformation and microstructure of hairtail after thawing, compared with air thawing (AT) and RT”. The sentence needs modification.

Response 2: We have revised this problem in Line 18.

Comment 3: Line 24; …… chewiness (5.75 mJ) should be ……… chewiness (5.75 mJ) values.

Response 3: We have corrected the grammar error in this paragraph in Line 24.

Comment 4: Abstract: Line 27; These results of…..

Response 4: We have revised this sentence in Line 27.

Comment 5: Introduction: Lines 44; The authors did not perform any tests for lipid quality evaluation. So, I recommend to add any analysis for lipid oxidation tests.

Response 5: We have supplied the TBARs as the index representing the degree of lipid oxidation in Line 48.

Comment 6: Introduction: Lines 58-59; how does traditional thawing cause overcooking? Please clarify…

Response 6: We have clarified the reasons for over-ripening due to traditional thawing in the introduction according to the revision suggestions in Line 52.

Comment 7: Introduction: Lines 46-47; “Nevertheless, these methods have shown drawbacks, such as being time-consuming, uneven thawing, and inducing irreversible physical or chemical damage to fish products”. Please add reference.

Response 7: We have added more literature to support the disadvantage of thawing ways, and we referred to Ref. [8] in the revised manuscript, as seen in Line 66.

Comment 8: Introduction: Line 90; the word “realize” is not suitable to express meaning….

Response 8: We have changed the word to “achieve” in Line 97.

Comment 9: Introduction: Line 93-94; “The results obtained in this work will provide a theoretical basis for hairtail thawing treatment in the food industry” replace the sentence in conclusion.

Response 9: We have used new sentences to replace previous sentences in Line 100.

Comment 10: Sample Preparation: Line 97; Mention the length and weight of the fish.

Response 10: We have added the information that the length is approximately 60 cm, and each one weighs 0.5 kg of each hairtail (Trichiurus lepturus) in Line 105.

Comment 11: Sample Preparation: Line 100; Why did the authors freeze at –80 °C? Usual freezing is done at –40 °C.

Response 11: We have referred to other storage method as Ref. [19] in the revised manuscript, as seen in Line 108.

Comment 12: Experimental Setup: Line 103; The authors should state the source/ production device of oxidized graphene nanoparticles. They also require to add the instrumental model for a radio frequency thawing device.

Response 12: We have supplied more specific information in Line 113.

Comment 13: Name the device model and origin used to measure center temperature.

Response 13: We have named the device model in Line 123.

Comment 14: Texture Analysis: Mention full specification of texture analyzer.

Response 14: We have supplied the information about the full specification of texture analyzer in Line 127.

Comment 15: Measurement of Particle Size: Add reference for the method used.

Response 15: We have added a reference for the method used in Line 201.

Comment 16: Results: The Figure 2 & 3 need quality improvement for the readers.

Response 16: We have modified the Figure 2 and 3.

Comment 17: Conclusion: State the limitations (if any) and future research suggestions.

Response 17: We have added the limitations and future research suggestions in Line 362.

Round 2

Reviewer 2 Report

Comments and Suggestions for Authors

Thank you very much for improving the manuscript.

Author Response

Thank you very much for reviewing our manuscript.